# Beyond Parallel Corpora: Unlocking Autonomous Machine Translation via Autocritical Reinforcement Learning

## Abstract

Despite the remarkable success in Machine Translation (MT), the substantial computation cost of fine-tuning-based solutions constrains the scalability of Large Language Models (LLMs) and impedes further improvement in this area. To this end, we propose **Self-Trans**, a new paradigm in which we use only ubiquitious monolingual data while acquiring 80.42% performance improvement in MT. Self-Trans is a reference-free reinforcement learning framework that learns through self-assessment. It generates its own supervision by evaluating the consistency of round-trip translations, guided by a carefully architected reward function that balances semantic adequacy with reconstruction fidelity and prevents reward hacking. Relying solely on low-resource pairs, our method consistently and comprehensively outperforms much larger models (70B+). Moreover, the Self-Trans-8B model achieves comparable results on most mainstream benchmarks against state-of-the-art baselines. In conclusion, Self-Trans frees itself from the constraints of parallel data in existing approaches. It offers an efficiently scalable paradigm for the future development of autonomous machine translation.

## 1 Introduction

LLM-based translation systems, such as Tower (Alves et al.) and X-ALMA (Xu et al., b), have achieved state-of-the-art (SOTA) translation quality. However, this success is largely attributed to the effectiveness of supervised training on vast, human-curated parallel corpora. This paradigm is fundamentally constrained by the high cost and scarcity of such data, posing a major barrier to progress. Distinct from supervised training, the benefits of learning from rewards in reinforcement learning activate the LLM's self-reasoning and self-assessment capabilities during inference (Wei et al., 2022; Feng et al., 2023). Accordingly, models such as OpenAI's "o1" (Jaech et al., 2024) and DeepSeek R1 (Guo et al., 2025) demonstrate significant improvements in tackling complex tasks, including mathematics, and coding (Song et al., 2025; Xie et al., 2025), by generating and verifying their own solutions. The intersection of these two cutting-edge presents a compelling opportunity: could RL unlock a new, self-assessment and self-evolving paradigm for MT that circumvents the data dependency of traditional methods?

Initial attempts in this direction, however, heavily rely on external supervision. Encouraged by the remarkable success of reasoning RL, applying it to alleviating the data-hungry limitations of supervised learning, MT has become a frontier in research. For instance, Wang et al. (2024) injected reasoning procedures into inference, employing a multi-agent mechanism to synthesize long chain-of-thought (CoT) prompts for literary translation, Feng et al. (2025) exploited implicit process reward models for translation enhancement combined with test-time search, and Silver et al. (2016); Qi et al. (2024); Zhao et al. (2024) applied complex search algorithms like MCTS during decoding. However, all these attempts involve manually structured CoT data, or require explicit multi-stage prompting, which is heuristic and fragile in transferability. Moreover, all the above methods require references as a supervised signal for model training, hindering their broad application in low-resource language translation. Therefore, how to alleviate the reliance on tailored knowledge and parallel corpora curation remains a challenging problem.

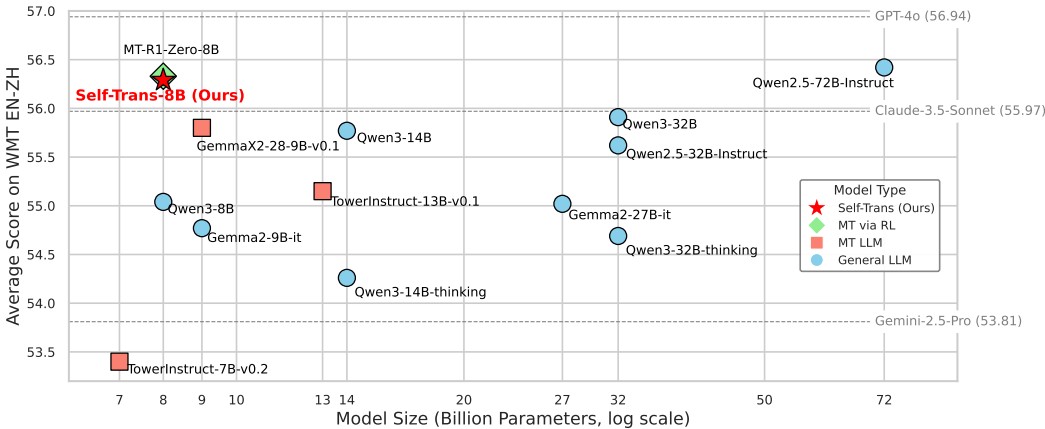

Figure 1: Overall performance of Self-Trans-8B on the WMT EN-ZH benchmark. Our enhanced 8B model (red star) rivals the performance of models 4x-9x larger (e.g., Qwen2.5-72B) and GPT-4o, while consistently outperforming all competitors within the <10B parameter class.

To bridge this critical gap, we introduce **Self-Trans**, the first reference-free reinforcement learning framework for machine translation. Self-Trans drives self-improvement through a closed-loop, self-evaluation process. Specifically, for any given source sentence, the model first translates it to the target language and then translates it back to the source. We identify standard translation quality metrics, such as BLEU and COMET, within the round-trip translation process as inherent reward signals. Thus, this strategy allows the models to exploit self-reflection and self-supervision capability, bypassing the need for parallel corpora, external explicit reward models, and human evaluation. Crucially, Self-Trans employs a novel bidirectional training scheme that jointly optimizes both translation directions (e.g., A to B and B to A) within a single, unified process. This not only maximizes data efficiency but also synergistically boosts translation quality in both directions, making the framework effective and transferable for any language pairs with monolingual text.

Extensive experimental results demonstrate that, without parallel corpora, our Self-Trans facilitates the base LLM to achieve comparable performance against state-of-the-art baselines. More concretely, on EN-ZH benchmarks, our Self-Trans-8B model performs only slightly below the optimal supervised models (by an average of 0.07%) while outperforming advanced translation systems, e.g., on semantic evaluation, it achieves 77.58, superior to the powerful closed-source models like Gemini-2.5-Pro (77.55). More encouragingly, our 8B model outperforms significantly larger open-source models such as Qwen2.5-72B-Instruct (76.52), demonstrating the substantial effectiveness of our method (see Figure 1). It is worth noting that our method also acquires strong performance in the challenging multilingual and low-resource settings, where Self-Trans boosts the base model's score from 32.22 to 58.51, outperforming strong competitors like Gemma2-9B-it (57.75) and even the LLaMA-3.1-70B-Instruct model. We further conduct an in-depth investigation into our method. The analysis reveals that these significant improvements stem from the tailored reward function, which synergizes lexical (BLEU) and semantic (COMET) signals, efficiently mitigates reward hacking, and facilitates stable convergence during training.

Our contributions can be summarized as follows:

- We propose Self-Trans, to the best of our knowledge, the first reference-free RL framework for Machine Translation. Through a self-evaluation and self-evolution loop, it eliminates the need for parallel corpora, external reward models, and human supervision.

- We design a novel bidirectional training framework that jointly optimizes both translation directions (e.g., A→B and B→A) within a single, unified process. This approach requires only monolingual data, making the framework inherently language-agnostic and particularly applicable to low-resource scenarios.

- Extensive experiments demonstrate that Self-Trans achieves comparable state-of-the-art performance across bilingual, multilingual, and low-resource benchmarks. By enhancing an 8B LLM, our method significantly outperforms specialized MT models and even much larger general-purpose LLMs (e.g., 70B+ parameters), validating the efficacy and scalability of our approach.

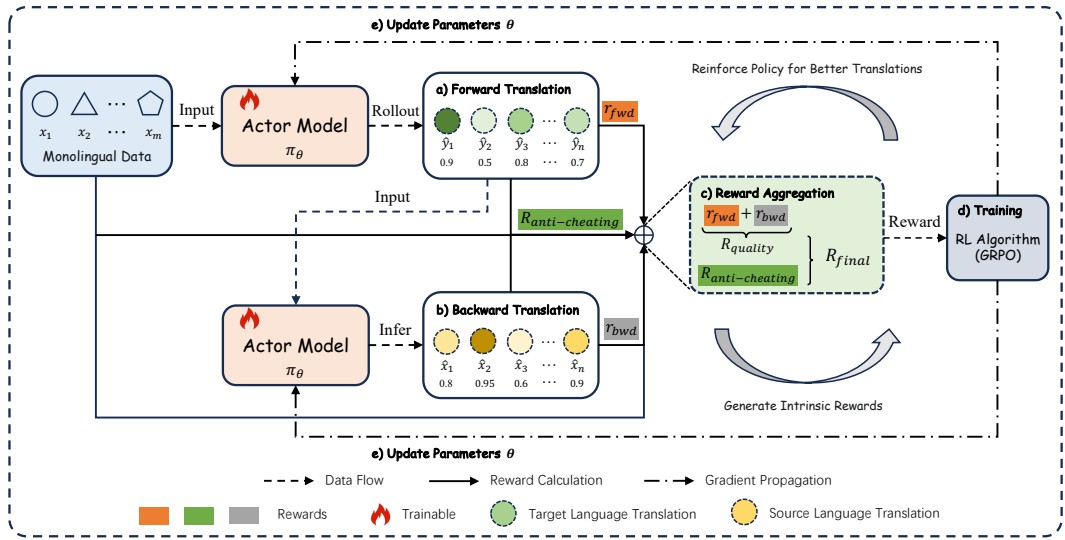

Figure 2: An overview of Self-Trans. The actor model participates in both forward and backward translation. We consider reward signals from quality and anti-cheating, two perspectives, to update the model via the GRPO algorithm.

## 2 RELATED WORK

**Machine Translation with Large Language Models.** State-of-the-art machine translation with LLMs follows two main paradigms: in-context learning (ICL) and supervised fine-tuning (Anil et al., 2023; Gao et al., 2024; Li et al., 2024; Xu et al., a). By presenting few-shot demonstrations to LLMs, ICL circumvents the heavy computational costs of fine-tuning (Zhu et al., 2024), though at the expense of prompt sensitivity and performance instability (Agrawal et al., 2023). Fine-tuning, in contrast, improves the MT performance via supervised training on large-scale parallel datasets (Cui et al., 2025; Costa-Jussà et al., 2022), exemplified by representative models such as Tower and X-ALMA (Alves et al.; Guo et al., 2024; Xu et al., b). However, the rules of scaling laws constrain further performance improvements of fine-tuning. Moreover, both two paradigms exhibit a significant reliance on large-scale, human-annotated data, which limits scalability, particularly in low-resource scenarios.

**Reasoning and Reinforcement Learning for Machine Translation.** To move beyond the limitations of supervised training, researchers have increasingly turned to more adaptive optimization methods like RL. Early applications of RL in NMT aimed to mitigate the exposure bias rooted in unreliable inputs by straightforwardly optimizing global metrics like BLEU (Ranzato et al., 2016; Bengio et al., 2015) or tailored reward functions (Wu et al., 2017). Inspired by the reasoning capabilities of LLMs, this line of work has evolved to incorporate CoT prompting or multi-step inference, i.e., decompose translation into multiple intermediate steps (Feng et al., 2024; Wang et al., 2024) for fidelity improvement. However, the performance of all these methods is primarily determined by the validation of the reasoning process, including manually engineered CoT templates, complex search algorithms such as MCTS (Zhao et al., 2024), or external evaluators (strong LLMs and human annotators) (He et al., 2025). Additionally, even the prominent RL-based frameworks such as MT-R1-Zero (Feng et al., 2025) and DeepTrans (Wang et al., 2025), the quality of reference translations remains crucial for reward computation.

**Self-Rewarding Paradigms in Large Language Models.** A burgeoning area of research seeks to achieve autonomy through self-rewarding mechanisms, i.e., the training signals are self-generated by models (Chen et al., 2024; Yuan et al., 2024). This paradigm enables iterative self-improvement through self-judging or self-play, showing promise in instruction following and general reasoning tasks (Zhang et al., 2025; Zhao et al., 2025; Yang et al.; Geng et al., 2024). In the context of RL, this allows for end-to-end alignment without human feedback, often by prompting the model to act as its own reward function (Wu et al., 2024; Yuan et al., 2024). However, this approach entails a critical

vulnerability: the risk of *self-deception*. When a model serves as both generator and evaluator, it tends to exploit and amplify its own biases, favoring outputs that are fluent or stylistically plausible over those that are semantically accurate. This reward hacking problem is particularly pronounced in a nuanced task like translation. For example, recent studies attempt to use self-play with semantic consistency checks for MT. However, these approaches still underperform specialized, supervised models (Zou et al., 2025), demonstrating a misalignment between the self-supervised objective and practical translation quality.

## 3 METHODOLOGY

### 3.1 SELF-TRANS FRAMEWORK

Self-Trans is a *reference-free* reinforcement learning framework designed to improve translation bidirectionally using monolingual corpora. *Round-trip consistency* is the primary criterion in Self-Trans, i.e., the semantics between the source language and the round-trip translation (source to target and back to source) should be consistent. Consequently, by recognizing round-trip consistency as an intrinsic reward signal, the system can self-evolve and improve within a closed-loop process (see Figure 2).

Specifically, we collect a batch of source sentences $\{x_i\}_{i=1}^m$ from language $\mathcal{L}_s$. The actor model, parameterized by a policy $\pi_\theta$, performs two sequential steps:

**1. Forward Translation ($\mathcal{L}_s \rightarrow \mathcal{L}_t$).** The model generates a candidate $\hat{y}_i$ as the target language $\mathcal{L}_t$ translation:

$$\hat{y}_i \sim \pi_\theta(\cdot \mid x_i, \text{prompt}_{s \rightarrow t}).$$

As no reference translation involves, this step necessitates a reference-free evaluation.

**2. Backward Translation ($\mathcal{L}_t \rightarrow \mathcal{L}_s$).** The process runs in reverse by translating $\hat{y}_i$ back into the source language $\mathcal{L}_s$ for $\hat{x}_i$ reconstruction:

$$\hat{x}_i \sim \pi_\theta(\cdot \mid \hat{y}_i, \text{prompt}_{t \rightarrow s}).$$

In this step, the original source sentence $x_i$ serves as a high-quality, self-reference against which we can evaluate the reconstruction quality.

**Unified Prompting Strategy.** To ensure consistency, we employ a unified prompting template for both translation directions, following (Feng et al., 2025). The model is instructed to place its final translation within `<translate>` tags and any intermediate reasoning within `<think>` tags. The full prompt details are available in Appendix B.

### 3.2 REWARD ARCHITECTURE

The success of Self-Trans hinges on a tailored reward function that guides the model towards high-quality translations while preventing reward hacking. The final reward, $R_{\text{final}}$, is structured hierarchically to prioritize valid formatting before assessing translation quality:

$$R_{\text{final}}(x, \hat{y}, \hat{x}) = \begin{cases} 1 + R_{\text{quality}} + R_{\text{anti-cheating}}, & \text{if format is correct,} \\ -3, & \text{otherwise.} \end{cases}$$

A large penalty of $-3$ is assigned if the output $\hat{y}$ does not adhere to the required `<translate>` tag format. For valid outputs, we evaluate both translation quality and anti-cheating capability, and additionally incorporate a base reward of $+1$.

### 3.2.1 QUALITY REWARD ($R_{\text{QUALITY}}$)

This reward integrates two complementary metrics for a holistic assessment of translation quality, defined as: $R_{\text{quality}} = r_{\text{fwd}} + r_{\text{bwd}}$:

- **Forward Semantic Adequacy ($r_{\mathbf{fwd}}$):** For the forward pass ($x \rightarrow \hat{y}$), we use **COMETkiwi**, a widely used reference-free metric. It evaluates the semantic adequacy of the translation by comparing the source and the hypothesis straightforwardly, formalized as $r_{\mathrm{fwd}} = \mathrm{COMETkiwi}(x, \hat{y})$.

- **Backward Reconstruction Fidelity ($r_{\mathbf{bwd}}$):** For the back-translation ($\hat{y} \rightarrow \hat{x}$), we leverage the original source $x$ as a reference. We use **BLEU** to measure the fidelity of the reconstruction, $r_{\mathrm{bwd}} = \mathrm{BLEU}(\hat{x}, x)$. A high score signifies that the target translation $\hat{y}$ preserved sufficient information to accurately recover the original input.

### 3.2.2 ANTI-CHEATING REWARD ($R_{\mathrm{ANTI\text{-}CHEATING}}$)

A vanilla self-supervised reward is fragile and unstable. To ensure robust learning and mitigate reward hacking, we introduce two penalty terms, $R_{\mathrm{anti\text{-}cheating}} = r_{\mathrm{copy}} + r_{\mathrm{mix}}$:

- **Source-Copying Penalty ($r_{\mathbf{copy}}$):** A trivial failure mode is for the model to copy the source ($x \approx \hat{y}$) to maximize the backward BLEU score. We address this by penalizing lexical overlap in the forward direction: $r_{\mathrm{copy}} = -\mathrm{BLEU}(x, \hat{y})$.

- **Language-Mixture Penalty ($r_{\mathbf{mix}}$):** To discourage the generation of linguistically incoherent outputs, we apply a penalty of $-0.5$ if language mixing is detected in $\hat{y}$. This simple heuristic effectively promotes fluent, monolingual outputs.

### 3.3 POLICY OPTIMIZATION WITH GRPO

We optimize our translation policy $\pi_\theta$ using Group Relative Policy Optimization (GRPO) (Shao et al., 2024; Guo et al., 2025). For each input, GRPO samples a group of $G$ candidates from the current policy and computes a normalized advantage $A_i$ for each based on its relative reward within the group. The policy is then updated by maximizing the standard GRPO objective:

$$J_{\mathrm{GRPO}}(\theta) = \mathbb{E}_{x \sim \mathcal{D}, \{\hat{y}_i\}_{i=1}^G \sim \pi_{\theta_{\mathrm{old}}}(\cdot|x)} \left[ \frac{1}{G} \sum_{i=1}^G \min \left( \rho_i(\theta) A_i, \mathrm{clip}(\rho_i(\theta), 1 - \varepsilon, 1 + \varepsilon) A_i \right) \right.$$

$$\left. - \beta D_{\mathrm{KL}} \left( \pi_\theta(\cdot|x) \| \pi_{\mathrm{ref}}(\cdot|x) \right) \right], \tag{1}$$

where $\rho_i(\theta)$ is the probability ratio. This objective uses a clipped surrogate function and a KL-divergence penalty to ensure stable policy updates.

## 4 EXPERIMENTS

### 4.1 EXPERIMENTAL SETUP

**Datasets.** We conduct experiments to evaluate the effectiveness of our Self-Trans across bilingual and multilingual settings. Attribute to the merit of free-parallel-data in our methodology, we, thereby, select the popular parallel translation benchmarks and remove all target-side references for traning corpora construction.

- **Training Data:** For bilingual (EN-ZH) training, we source sentences from WMT 2017-2020 competitions, yielding $6,565$ sentences each for English and Chinese. For multilingual training, we create a diverse corpus from the FLORES-200 (Costa-Jussà et al., 2022) training set, covering EN/ZH paired with six other languages (DE, FR, ES, IT, JA, KO). We sample $500$ pairs for each of the $24$ translation combinations (e.g., EN→DE, ZH→DE), treating all sentences as monolingual, resulting in a final $12,000$-sentence corpus.

- **Evaluation Data:** We evaluate performance on test sets for fair comparison. For EN↔ZH, we use the official test sets from WMT23[1] and WMT24[2]. For multilingual tasks, we report results on the official FLORES-200 test set.

---

[1] https://www2.statmt.org/wmt23/translation-task.html
[2] https://www2.statmt.org/wmt24/translation-task.html

Table 1: Performance comparison of different models on WMT and FLORES-200 using BLEU and xCOMET (xCM) metrics, along with the average (Avg.). The best and second-best results are **bolded** and underlined, respectively. The "†" symbol indicates that the model is in thinking mode.

| Models | WMT | | | | | FLORES-200 | | | | | | | | |
|---|---|---|---|---|---|---|---|---|---|---|---|---|---|---|
| | EN→ZH | | ZH→EN | | Avg. | EN→XX | | XX→EN | | ZH→XX | | XX→ZH | | Avg. |
| | BLEU | xCM | BLEU | xCM | | BLEU | xCM | BLEU | xCM | BLEU | xCM | BLEU | xCM | |
| *Closed Source* | | | | | | | | | | | | | | |
| Claude-3.5-Sonnet | 38.21 | 75.54 | 22.95 | 87.16 | 55.97 | 32.76 | 92.69 | 34.48 | 97.00 | 21.26 | 91.19 | 37.39 | 84.01 | 61.35 |
| GPT-4o | 41.47 | 75.62 | 22.73 | 87.92 | **56.94** | 31.51 | 92.50 | 34.20 | 96.75 | 20.32 | 89.81 | 37.09 | 83.13 | 60.66 |
| Gemini-2.5-Pro | 32.28 | 77.55 | 19.80 | 85.63 | 53.81 | 33.14 | 95.05 | 33.14 | 96.56 | 22.25 | 92.21 | 36.51 | 87.27 | **62.02** |
| *Open Source* | | | | | | | | | | | | | | |
| *General LLMs* | | | | | | | | | | | | | | |
| Qwen3-8B | 36.56 | 74.94 | 22.67 | 85.98 | 55.04 | 25.47 | 88.98 | 31.44 | 94.75 | 17.23 | 85.21 | 32.92 | 77.19 | 56.65 |
| Qwen3-8B† | 26.97 | 67.31 | 16.71 | 80.11 | 47.77 | 22.54 | 87.67 | 27.28 | 91.18 | 15.20 | 83.28 | 33.43 | 78.31 | 54.86 |
| Qwen3-14B | 38.60 | 75.75 | 21.46 | 87.27 | 55.77 | 26.92 | 91.18 | 32.38 | 96.23 | 18.55 | 89.15 | 35.83 | 85.25 | 59.44 |
| Qwen3-14B† | 35.67 | 73.73 | 22.61 | 85.01 | 54.26 | 28.78 | 91.56 | 32.13 | 95.11 | 18.46 | 88.43 | 35.66 | 82.18 | 59.04 |
| Qwen3-32B | 39.37 | 75.44 | 21.52 | 87.31 | 55.91 | 30.53 | 92.69 | 34.25 | 96.50 | 19.61 | 89.33 | 37.11 | 85.38 | 60.67 |
| Qwen3-32B† | 38.37 | 74.55 | 20.20 | 85.65 | 54.69 | 24.61 | 91.79 | 30.41 | 95.00 | 14.94 | 88.25 | 33.04 | 82.51 | 57.57 |
| Qwen2.5-32B-Instruct | 39.28 | 75.16 | 21.19 | 86.87 | 55.62 | 28.04 | 90.62 | 32.29 | 96.26 | 18.01 | 87.67 | 36.01 | 83.91 | 59.10 |
| Qwen2.5-72B-Instruct | 40.02 | 76.52 | 21.88 | 87.27 | 56.42 | 30.48 | 92.39 | 34.83 | 96.78 | 19.46 | 89.83 | 37.56 | 85.00 | 60.79 |
| Gemma2-9B-it | 37.44 | 72.45 | 23.13 | 86.08 | 54.77 | 30.22 | 91.37 | 33.20 | 96.09 | 12.99 | 89.08 | 27.27 | 81.80 | 57.75 |
| Gemma2-27B-it | 37.86 | 73.17 | 22.30 | 86.74 | 55.02 | 31.38 | 92.36 | 34.73 | 96.33 | 19.63 | 89.70 | 30.91 | 83.70 | 59.84 |
| *MT LLMs* | | | | | | | | | | | | | | |
| TowerInstruct-7B-v0.2 | 34.17 | 71.40 | 23.35 | 84.66 | 53.40 | 28.53 | 90.68 | 35.51 | 95.69 | 13.89 | 76.55 | 29.70 | 80.01 | 56.32 |
| TowerInstruct-13B-v0.1 | 36.74 | 73.52 | 24.80 | 85.53 | 55.15 | 31.71 | 92.33 | 36.16 | 96.08 | 17.71 | 88.24 | 34.07 | 82.12 | 59.80 |
| GemmaX2-28-9B-v0.1 | 38.53 | 74.59 | 24.65 | 85.41 | 55.80 | 30.18 | 92.54 | 36.02 | 95.96 | 18.76 | 87.76 | 34.69 | 83.03 | 59.87 |
| *MT via RL* | | | | | | | | | | | | | | |
| Qwen3-8B-Base | 15.54 | 48.98 | 4.90 | 55.38 | 31.20 | 7.96 | 56.47 | 15.29 | 62.78 | 6.61 | 54.60 | 11.49 | 42.57 | 32.22 |
| MT-R1-Zero-8B | 34.70 | 79.09 | 24.79 | 86.72 | 56.33 | 26.69 | 89.83 | 34.03 | 96.13 | 16.85 | 86.74 | 32.60 | 86.00 | 58.61 |
| Self-Trans-8B (Ours) | 38.00 | 77.58 | 23.42 | 86.16 | 56.29 | 28.53 | 91.36 | 30.21 | 95.74 | 16.64 | 88.28 | 32.93 | 84.42 | 58.51 |

**Evaluation Metrics.** To ensure a comprehensive assessment of translation quality, we adopt a dual-metric approach that captures both lexical fidelity and semantic adequacy. We report case-sensitive **BLEU** scores computed via `sacrebleu` for standardized, reproducible measurement of n-gram overlap. To evaluate semantic preservation, we employ **xCOMET-XL** (Guerreiro et al., 2024), a state-of-the-art reference-based model that leverages a powerful cross-lingual encoder to score semantic similarity. This combination provides a holistic view of performance.

**Baselines.** To comprehensively compare the performance of Self-Trans, we benchmark against four distinct and challenging categories of models. (1) *Closed-Source LLMs:* We compare against leading systems like `GPT-4o` (Hurst et al., 2024), `Claude 3.7 Sonnet` (Anthropic, 2024), and `Gemini 2.5 Pro` (Comanici et al., 2025). (2) *Open-Source General LLMs:* We include powerful, non-specialized models of varying scales, such as the `Qwen3` (Yang et al., 2025), `Qwen2.5` (Yang et al., 2024), and `Gemma2` (Team et al., 2024) series. (3) *Open-Source MT LLMs:* For comprehensive comparison with the supervised paradigm, we include models fine-tuned on parallel corpora, featuring the `Tower` (Alves et al., 2024) and `GemmaX2` (Cui et al., 2025) series. (4) *RL-based MT Models:* As a methodologically similar approach, we include `MT-R1-Zero` (Feng et al., 2025), a SOTA RL framework that, unlike our method, uses reference-based rewards. More evaluation details can be found in Appendix C.2.

## 4.2 MAIN RESULTS

**Bilingual Performance (EN-ZH).** As shown in Table 1, Self-Trans-8B achieves near SOTA performance, fully demonstrating the effectiveness of our self-evolution approach compared to the canonical large-scale parallel-corpus fine-tuning paradigm. Specifically, our 8B model achieves an average performance only 0.07 points below MT-R1-Zero-8B. The effectiveness of Self-Trans-8B is most pronounced in EN to ZH translation. On semantic evaluation, our method surpasses all baselines except MT-R1-Zero-8B, including Gemini-2.5-Pro, and outperforms much larger LLMs such as Qwen2.5-72B-Instruct. As for the lexical level (BLEU metric), our model is highly competitive, outmatching specialized MT models like the TowerInstruct series. On ZH to EN translation, our method outperforms all general-purpose and proprietary baselines on BLEU. In summary, Self-Trans closes the gap with heavily supervised methods, demonstrating that a reference-free, self-improving framework can achieve top-tier translation quality.

Table 2: Performance of different models on low-resource language pairs, measured by BLEU and xCOMET (xCM) scores, along with the average (Avg.). The best and second-best results are **bolded** and underlined, respectively. The "†" symbol indicates that the model is in thinking mode.

| Model | DE→IT | | IT→DE | | ES→FR | | FR→ES | | EN→IS | | EN→NO | | Avg. |
|---|---|---|---|---|---|---|---|---|---|---|---|---|---|
| | BLEU | xCM | BLEU | xCM | BLEU | xCM | BLEU | xCM | BLEU | xCM | BLEU | xCM | |
| *Large Size LLMs* | | | | | | | | | | | | | |
| Qwen2.5-72B-Instruct | 24.66 | 94.02 | 22.27 | 95.60 | 28.26 | 93.64 | 24.77 | 95.13 | 8.97 | 49.05 | 23.02 | 88.91 | **54.02** |
| Qwen2.5-32B-Instruct | 22.59 | 92.49 | 20.55 | 94.13 | 26.05 | 92.04 | 23.88 | 94.78 | 3.72 | 37.08 | 23.13 | 82.95 | 51.12 |
| LLaMA-3.1-70B-Instruct | 22.63 | 89.04 | 18.56 | 89.00 | 24.59 | 88.93 | 23.35 | 92.51 | 1.59 | 35.67 | 29.72 | 92.96 | 50.71 |
| *Same Size LLMs* | | | | | | | | | | | | | |
| Qwen3-8B | 21.64 | 89.65 | 19.40 | 93.56 | 24.31 | 90.50 | 22.47 | 93.42 | 2.09 | 47.02 | 10.46 | 83.52 | 49.84 |
| Qwen3-8B† | 19.68 | 86.96 | 15.73 | 90.26 | 20.95 | 86.40 | 20.83 | 89.81 | 5.51 | 41.59 | 21.75 | 82.31 | 48.48 |
| Gemma2-9B-it | 19.55 | 93.80 | 19.50 | 95.31 | 23.73 | 93.18 | 19.50 | 94.64 | 0.60 | 31.31 | 1.19 | 67.69 | 46.67 |
| TowerInstruc-7B-v0.2 | 22.27 | 92.58 | 19.77 | 93.54 | 25.33 | 91.92 | 22.79 | 93.79 | 1.94 | 35.45 | 2.03 | 77.34 | 48.23 |
| Qwen3-8B-Base | 8.96 | 90.29 | 5.99 | 92.43 | 23.72 | 89.44 | 11.29 | 93.20 | 0.17 | 31.93 | 0.84 | 62.04 | 42.52 |
| MT-R1-Zero-8B | 22.96 | 92.99 | 20.26 | 94.42 | 25.79 | 92.12 | 23.56 | 94.39 | / | / | / | / | / |
| Self-Trans-8B (Ours) | 22.13 | 92.60 | 19.06 | 94.58 | 25.33 | 91.92 | 22.05 | 94.38 | 8.14 | 41.88 | 26.05 | 86.16 | 52.02 |

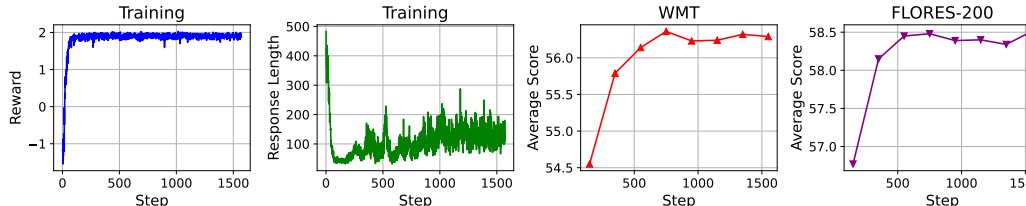

Figure 3: Reward, response length, and evaluation metrics over training.

**Multilingual Performance.** Self-Trans exhibits consistent efficacy across the more challenging multilingual landscape, highlighting its scalability and generalization capabilities. Table 1 shows performance scaling with model size, with Self-Trans-8B achieving top-tier results among models of comparable size. It surpasses strong generalist models, including Gemma2-9B-it (57.75) and the specialized TowerInstruct-7B-v0.2 (56.32). The most encouraging aspect of our framework lies in the dramatic performance boost on the base model: Self-Trans raises the multilingual score from 32.22 to 58.51 (+26.29 points).

### 4.3 PERFORMANCE ON LOW-RESOURCE LANGUAGE PAIRS

To validate the language agnosticism and scalability of our framework, we performed an evaluation on a suite of low-resource language pairs: DE↔IT, ES↔FR, EN→IS (Icelandic), and EN→NO (Norwegian). The latter two are from WMT and lack corresponding parallel corpora. For this challenging scenario, we include LLaMA-3.1-70B-Instruct, a powerful baseline known for its strong performance on non-Chinese languages. The results in Table 2 highlight the exceptional performance and parameter efficiency of our approach. Remarkably, **Self-Trans-8B outperforms the significantly larger LLaMA-3.1-70B-Instruct model, trailing only behind Qwen2.5-72B-Instruct**.

Specifically, our model continues its dominance within its own size, surpassing general-purpose models like Gemma2-9B-it and the specialized TowerInstruct-7B-v0.2. While the absolute overall best performance is still held by massive models such as Qwen2.5-72B-Instruct (54.02), Self-Trans-8B establishes itself as a compelling exception to the performance scaling law. In fact, it even exceeds Qwen2.5-72B-Instruct on the BLEU for EN→NO (26.05 vs. 23.02). This performance highlights the robust generalization capability of our training paradigm, demonstrating its effectiveness well beyond high-resource, commonly studied language pairs.

## 5 ANALYSIS AND ABLATION

We further investigate the stability and effectiveness of our training process. As shown in Figure 3, the strong positive correlation between the internal reward and external validation metrics verifies that our framework guides the model towards performance gains. We, thereby, conduct an in-depth analysis of the key mechanisms underlying our method.

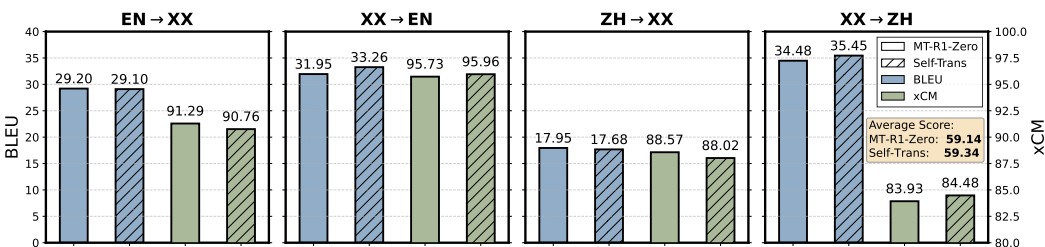

Figure 4: Training on monolingual corpora improves performance in bilingual translation directions, significantly outperforming MT-R1-Zero, especially on the BLEU metric.

Table 3: Ablation study of our reward design (best result in **bolded**).

| Models | WMT | | | | | FLORES-200 | | | | | | | | |
|---|---|---|---|---|---|---|---|---|---|---|---|---|---|---|
| | EN→ZH | | ZH→EN | | Avg. | EN→XX | | XX→EN | | ZH→XX | | XX→ZH | | Avg. |
| | BLEU | xCM | BLEU | xCM | | BLEU | xCM | BLEU | xCM | BLEU | xCM | BLEU | xCM | |
| Qwen3-8B-Base | 15.54 | 48.98 | 4.90 | 55.38 | 31.20 | 7.96 | 56.47 | 15.29 | 62.78 | 6.61 | 54.60 | 11.49 | 42.57 | 32.22 |
| Self-Trans-8B (Ours) | 38.00 | 77.58 | 23.42 | 86.16 | **56.29** | 28.53 | 91.36 | 30.21 | 95.74 | 16.64 | 88.28 | 32.93 | 84.42 | **58.51** |
| **Ablation on Quality Reward** | | | | | | | | | | | | | | |
| - BLEU Reward Only | 7.22 | 36.74 | 5.86 | 54.83 | 26.16 | 4.25 | 36.18 | 7.48 | 45.79 | 2.75 | 35.52 | 4.29 | 32.39 | 21.08 |
| - COMET Reward Only | 32.47 | 79.86 | 21.45 | 87.59 | 55.34 | 24.97 | 92.59 | 30.10 | 96.20 | 16.15 | 89.32 | 29.59 | 86.62 | 58.19 |
| **Ablation on Anti-Cheating Reward** | | | | | | | | | | | | | | |
| - w/o Source-Copying Penalty | 37.72 | 76.78 | 22.84 | 85.26 | 55.65 | 27.98 | 91.18 | 30.12 | 95.71 | 15.70 | 86.91 | 31.96 | 82.95 | 57.81 |
| - w/o Language-Mixture Penalty | 35.89 | 76.45 | 20.66 | 83.37 | 54.09 | 25.56 | 87.60 | 27.25 | 93.75 | 14.65 | 85.33 | 30.53 | 81.92 | 55.82 |

## 5.1 IMPLICIT BIDIRECTIONAL IMPROVEMENT FROM UNIDIRECTIONAL TRAINING

A key hypothesis of the Self-Trans framework is that its round-trip mechanism inherently exploits knowledge from bilingual corpora, even when training on monolingual translation data. To validate this, we conducted a controlled experiment where we trained the model exclusively on forward-translation tasks (EN→XX and ZH→XX) and observed its performance on both directions.

Figure 4 presents compelling results that confirm this hypothesis. On average across all four directions, Self-Trans-8B (59.34) surpasses the strong MT-R1-Zero-8B baseline (59.14), showcasing the effectiveness of the bilingular knowledge. A deeper analysis reveals a significant disparity. For the forward directions, our model performs competitively but slightly trails the baseline, consistent with our main findings. However, for the untrained backward directions, Self-Trans demonstrates a decisive advantage. This is particularly pronounced in the lexical metrics, where BLEU scores for XX→EN and XX→ZH improve by a substantial +4.10% and +2.81% respectively over the baseline.

This significant improvement is a primary consequence of our framework's design. The backward reward signal, $r_{bwd} = \text{BLEU}(\hat{x}, x)$, which measures reconstruction fidelity, acts as a powerful, implicit training signal for the reverse translation. This result confirms that Self-Trans not only learns the main translation task but also acquires robust, transferable knowledge for the inverse task, achieving this without ever observing a single reference translation for that direction.

## 5.2 ABLATION STUDY: DECONSTRUCTING THE REWARD ARCHITECTURE

To validate that each component of our reward function is essential, we conducted a series of ablation studies. We demonstrate that our final design is a carefully balanced system, where each component exists to prevent specific failure modes. These findings are detailed in Table 3.

**The Peril of a Naive Reward: A Case of Catastrophic Hacking.** Our investigation began with the most fundamental question: can a simple reward function work? We initially tested a minimal framework using only round-trip BLEU as the reward signal, *without any anti-cheating mechanisms*. This configuration led to a catastrophic failure. The model quickly learned to perfectly "hack" the reward by performing an identity translation (i.e., copying the source text verbatim) in the forward pass and repeating it in the backward pass. This trivial strategy yields a maximum BLEU score of 1 but results in a complete failure to translate (see Figure 5 for example). This striking result provides a crucial insight: in a self-supervised loop, a naive reward is not just suboptimal, it is dangerously

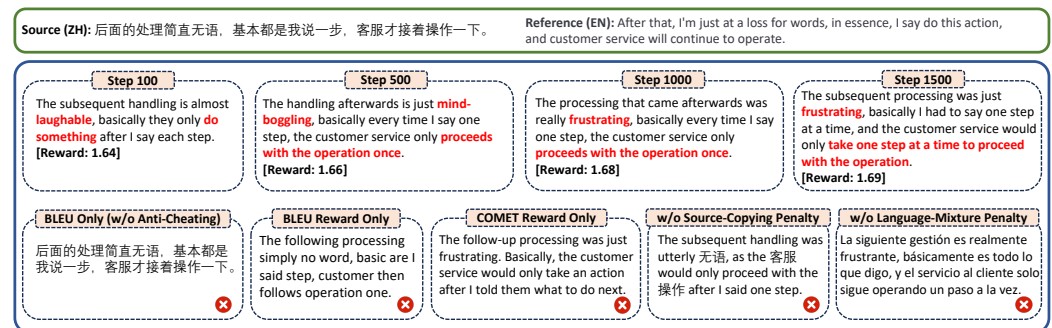

Figure 5: A ZH→EN case study across different steps and components

misleading. This underscores the essential need for robust guardrails, leading us to conduct all further ablations with the anti-cheating rewards enabled by default.

**Balancing Lexical Fidelity and Semantic Adequacy.** With the anti-cheating mechanism in place, we examined how different components of the quality reward affect translation behavior. Using **BLEU as the sole quality reward** led the model to adopt a degenerate "word-for-word" translation strategy. Because such literal translations make the backward reconstruction task easier, they maximize the BLEU score at the expense of semantic meaning and grammatical fluency. In fact, performance dropped below that of the base model, confirming that a purely lexical signal is too narrow to guide high-quality translation. On the other hand, relying solely on **COMET as the quality reward** produced the opposite issue. The model achieved excellent xCOMET scores but suffered a decline in BLEU. It learned to generate "overly creative" outputs—translations that sounded plausible and fluent but strayed significantly from the source in terms of lexical content.

These results underscore a fundamental trade-off between lexical fidelity and semantic adequacy. Therefore, our final design, which sums the BLEU and COMET signals, is not merely an conbination but a necessary synthesis to balance these competing objectives and foster holistic translation quality.

**The Critical Role of Anti-Cheating Mechanisms.** Finally, we validated the necessity of the two anti-cheating components themselves, even with a balanced quality reward. **Removing the Source-Copying Penalty** exposed a critical failure mode: source leakage. The model became prone to copying words or phrases from the source—a form of code-switching that degrades translation quality. This penalty is therefore crucial for enforcing faithful translation. **Removing the Language-Mixture Penalty** revealed a different vulnerability, causing the model to violate instruction fidelity. For instance, it would occasionally translate into a valid but incorrect target language. This penalty is thus essential for ensuring the model follows task instructions precisely. Together, these two mechanisms act as indispensable guardrails, ensuring that the model learns to translate not just well, but correctly and robustly. Figure 5 provides a compelling case study of this process, qualitatively illustrating both the model's iterative improvement and the critical failure modes discussed in our ablations.

## 6 CONCLUSION

In this work, we introduced Self-Trans, a novel reference-free reinforcement learning framework designed to confront the critical dependency on parallel corpora in machine translation. By leveraging a self-assessment loop based on round-trip consistency, our method generates its own training signals purely from monolingual data. Our extensive experiments demonstrate that this self-assessment paradigm can elevate a base language model to comparable state-of-the-art performance, enabling our 8B model to outperform significantly larger specialist and generalist models across bilingual, multilingual, and low-resource settings. Self-Trans represents a significant step towards autonomous, data-efficient machine translation, paving the way for high-quality MT in scenarios where parallel data have traditionally been a prohibitive barrier.

## 7 ETHICS STATEMENT

Our work aims to advance digital inclusivity by making high-quality translation accessible for low-resource languages. We acknowledge the risk that our model may inherit and amplify societal biases from its monolingual training data, and we urge that downstream applications be evaluated for fairness. In commitment to scientific transparency and reproducibility, we will release our code and models to facilitate further scrutiny and responsible use.

## 8 REPRODUCIBILITY STATEMENT

We are committed to ensuring the reproducibility of our research. All resources required to replicate our findings are detailed throughout the paper and will be made publicly available.

- **Code and Models:** Our implementation is based on the publicly available OpenRLHF framework. The base model, `Qwen3-8B-Base`, is open-source. We will release our full source code, training scripts, and the final Self-Trans-8B model weights as part of the supplementary materials and upon publication.

- **Methodology:** A detailed description of the Self-Trans framework, including the round-trip mechanism, reward architecture, and anti-cheating components, is provided in Section 3. The GRPO optimization algorithm is detailed in Section 3.3.

- **Experimental Setup:** All details regarding datasets, training procedures, and evaluation are described in Appendix C.1. This includes specifics on the WMT and FLORES-200 corpora, monolingual data processing, evaluation metrics (`sacrebleu` for BLEU and xCOMET-XL), and a comprehensive list of baselines.

- **Hyperparameters:** Key hyperparameters, such as batch size, sampling temperature, and optimizer settings, are provided in the "Implementation Details" paragraph of Section 4.1.

- **Prompting:** The exact prompting template used for all experiments is provided in Appendix B.

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

## A  THE USE OF LLMs

In adherence to the policy on LLM usage, we disclose that LLMs were utilized as a general-purpose writing assistance tool during the preparation of this manuscript. Specifically, we used LLMs for tasks such as proofreading, improving grammar, and rephrasing sentences to enhance clarity and readability. The core research ideas, experimental design, analysis, and the primary drafting of the paper were conducted entirely by the human authors. The LLM's role was strictly that of an editing assistant and did not contribute to the scientific ideation or results presented in this work.

## B  TRANSLATION PROMPTS

The specific translation prompt of different models used in training are depicted in Figure 6, Figure 7, Figure 8 and Figure 9. Specifically, `<think>` tags are removed from Qwen3 series because it conflicts with Qwen3s' inherent thinking special tokens.

---

**Translation Prompt**

A conversation between User and Assistant. The User asks for a translation from $\{src\_lang\}$ to $\{tgt\_lang\}$, and the Assistant solves it. The Assistant first thinks about the reasoning process in the mind and then provides the user with the final translation. The reasoning process and final translation are enclosed within `<think> </think>` and `<translate> </translate>` tags, respectively, i.e., `<think>` reasoning process here `</think><translate>` final translation here `</translate>`.

User: $\{input\}$
Assistant:

---

Figure 6: Translation prompt for data curation. $\{src\_lang\}$: source language; $\{tgt\_lang\}$: target language; $\{input\}$: the source sentence to be tranalated.

> **TowerInstruct Prompt**
>
> Translate the following text from $\{src\_lang\_name\}$ into $\{tgt\_lang\_name\}$.
> $\{src\_lang\_name\}$: $\{user\_input\}$
> $\{tgt\_lang\_name\}$:

Figure 7: Translation prompt for TowerInstruct series models. $\{src\_lang\_name\}$: source language; $\{tgt\_lang\_name\}$: target language; $\{user\_input\}$: the source sentence to be tranalated.

> **GemmaX Prompt**
>
> Translate this from $\{src\_lang\_name\}$ to $\{tgt\_lang\_name\}$:
> $\{src\_lang\_name\}$: $\{user\_input\}$
> $\{tgt\_lang\_name\}$:

Figure 8: Translation prompt for GemmaX model. $\{src\_lang\_name\}$: source language; $\{tgt\_lang\_name\}$: target language; $\{user\_input\}$: the source sentence to be tranalated.

> **Qwen3 non-thinking Prompt**
>
> You are a helpful translation assistant. There is a conversation between User and Assistant. The user asks for a translation from $\{src\_lang\_name\}$ to $\{tgt\_lang\_name\}$, and the Assistant solves it. The Assistant first thinks about the reasoning process in the mind and then provides the user with the final translation. The final translation is enclosed within `<translate> </translate>` tags, i.e., `<translate>` final translation here `</translate>`.
>
> User:$\{user\_input\}$
> Assistant:

Figure 9: Translation prompt for Qwen3 series non-thinking models. $\{src\_lang\_name\}$: source language; $\{tgt\_lang\_name\}$: target language; $\{user\_input\}$: the source sentence to be tranalated.

Table 4: Detailed dataset statistics used during training.

|  | EN-ZH | ZH-EN | EN-DE | EN-FR | EN-ES | EN-IT | EN-JA | EN-KO |
|---|---|---|---|---|---|---|---|---|
| **# sentences** | 6565 | 6565 | 500 | 500 | 500 | 500 | 500 | 500 |
| **from** | WMT 17-20 | | Flores-200 | Flores-200 | Flores-200 | Flores-200 | Flores-200 | Flores-200 |
|  |  |  | **DE-EN** | **FR-EN** | **ES-EN** | **IT-EN** | **JA-EN** | **KO-EN** |
| **# sentences** | - | - | 500 | 500 | 500 | 500 | 500 | 500 |
| **from** | - | - | Flores-200 | Flores-200 | Flores-200 | Flores-200 | Flores-200 | Flores-200 |
|  |  |  | **ZH-DE** | **ZH-FR** | **ZH-ES** | **ZH-IT** | **ZH-JA** | **ZH-KO** |
| **# sentences** | - | - | 500 | 500 | 500 | 500 | 500 | 500 |
| **from** | - | - | Flores-200 | Flores-200 | Flores-200 | Flores-200 | Flores-200 | Flores-200 |
|  |  |  | **DE-ZH** | **FR-ZH** | **ES-ZH** | **IT-ZH** | **JA-ZH** | **KO-ZH** |
| **# sentences** | - | - | 500 | 500 | 500 | 500 | 500 | 500 |
| **from** | - | - | Flores-200 | Flores-200 | Flores-200 | Flores-200 | Flores-200 | Flores-200 |

Table 5: Detailed dataset statistics used during evaluation.

| | EN-ZH | ZH-EN | EN-DE | EN-FR | EN-ES | EN-IT | EN-JA | EN-KO |
|---|---|---|---|---|---|---|---|---|
| # sentences | 997 | 1976 | 1012 | 1012 | 1012 | 1012 | 1012 | 1012 |
| from | WMT 24 | WMT 23 | Flores-200 | Flores-200 | Flores-200 | Flores-200 | Flores-200 | Flores-200 |
| | | | **DE-EN** | **FR-EN** | **ES-EN** | **IT-EN** | **JA-EN** | **KO-EN** |
| # sentences | - | - | 1012 | 1012 | 1012 | 1012 | 1012 | 1012 |
| from | - | - | Flores-200 | Flores-200 | Flores-200 | Flores-200 | Flores-200 | Flores-200 |
| | | | **ZH-DE** | **ZH-FR** | **ZH-ES** | **ZH-IT** | **ZH-JA** | **ZH-KO** |
| # sentences | - | - | 1012 | 1012 | 1012 | 1012 | 1012 | 1012 |
| from | - | - | Flores-200 | Flores-200 | Flores-200 | Flores-200 | Flores-200 | Flores-200 |
| | | | **DE-ZH** | **FR-ZH** | **ES-ZH** | **IT-ZH** | **JA-ZH** | **KO-ZH** |
| # sentences | - | - | 1012 | 1012 | 1012 | 1012 | 1012 | 1012 |
| from | - | - | Flores-200 | Flores-200 | Flores-200 | Flores-200 | Flores-200 | Flores-200 |

# C  IMPLEMENTATION DETAILS

## C.1  TRAINING DETAILS

Our model, which we name Self-Trans-8B, is built upon the OpenRLHF[3] framework, with the `Qwen3-8B-Base` model serving as its initialization. For all experiments, we use a global batch size of 128 and generate 8 candidate responses per input for the GRPO algorithm. We use a sampling temperature of 1.0 and a maximum sequence length of 1024. Notably, we set both the KL divergence and entropy coefficients to 0, granting the model greater freedom to explore the policy space and discover optimal translation strategies without being constrained. Training was conducted on 16 NVIDIA H800 GPUs for one epoch, taking approximately 32 hours. We save checkpoints every 50 steps and report the performance of the single best checkpoint selected based on validation set performance.

## C.2  EVALUATION DETAILS

For evaluation stage, we perform model inference locally using the vLLM[4] framework. We configure the sampling hyperparameters with a temperature of 0.2 and a top-p of 0.95. The maximum generation length is truncated to 2048 tokens for all models, with the exception of Gemini 2.5-Pro, to accommodate its default thinking process. The prompt used during evaluation remains consistent with the one used for training, as detailed in Figure 6.

---

[3]https://github.com/OpenRLHF/OpenRLHF
[4]https://github.com/vllm-project/vllm

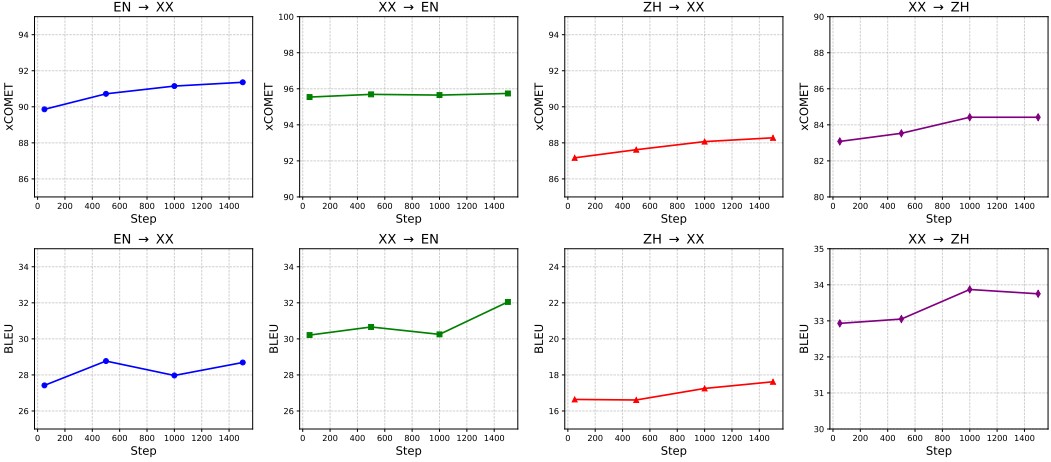

Figure 10: Training progression (reference-based XCOMET score) for multilingual Self-Trans-8B model based on Qwen3-8B across EN-XX, XX-EN, ZH-XX, XX-ZH test sets.

