# OpenReview forum: "Beyond Parallel Corpora: Unlocking Autonomous Machine Translation via Autocritical Reinforcement Learning"
_ICLR.cc/2026/Conference — Submitted to ICLR 2026_

### Official Review · Reviewer_Mkbq · 2025-10-28

**Soundness:** 1
**Presentation:** 2
**Contribution:** 1
**Rating:** 2
**Confidence:** 5

**Summary:**

This paper presents an unsupervised approach to machine translation (MT) specialization via GRPO with specialized rewards. The motivation is to replace fine-tuning on costly-to-create parallel data. The approach consists of a combination of rewards for forward and backward translation that combine two common MT evaluation metrics with language checks and a copy check. The approach is compared against various generalist multilingual LLMs of various sizes and a couple of MT-focused multilingual LLMs, on classic MT benchmarks sourced from WMT and Flores (subselection of languages) with BLEU and XComet-XL. RL with the new rewards lead to large improvements over the base model and yield competitive scores. The paper furthermore provides an ablation of the components of the reward.

While the idea and the execution are solid, the paper lacks novelty: The same approach of dual learning with RL was already proposed pre-LLMs in 2016 (Dual Learning for Machine Translation, He et al. 2016, NeurIPS). The main contribution of this paper is the more advanced crafting of rewards, but that is also due to novel metrics that did not exist in 2016. It also misses the entire literature on unsupervised MT (now coined "autonomous MT"). More importantly, it misses empirical comparisons to support its claim of more "efficiently scalable" approach than SFT, and independence of parallel translations and prevention of overfitting (details below) . The paper needs reframing and a more rigorous empirical evaluation. At the current stage it mostly demonstrates that it is not hard anymore to build competitive MT systems when the test domain is narrow and parallel data exists and metrics are known and progress is measured on the sentence level - but that is not novel, nor very surprising.

**Strengths:**

- The idea is nice and intuitive.
- The reward design seems to be well supported by experiments (ablations).
- Large list of baseline comparisons (albeit lacking a few core ones).
- Motivation to reduce resource-requirements is impactful (albeit not completely supported in experiments).

**Weaknesses:**

- Literature gaps: dual learning with RL for NMT (same idea, different RL algorithm because GRPO wasn’t around, https://proceedings.neurips.cc/paper_files/paper/2016/file/5b69b9cb83065d403869739ae7f0995e-Paper.pdf). Unsupervised machine translation (e.g. https://aclanthology.org/anthology-files/anthology-files/pdf/wmt/2020.wmt-1.68.pdf).
- The motivation doesn’t match the experimental setup: There is no comparable experiment with SFT (even though parallel data is used, so running the experiment is actually possible), and the low-resource reference-free setup is merely simulated on parallel data and predominantly high-resource language pairs (even though it would have been really easy to come up with a more realistic scenario, see my suggestions below).
- Efficiency of the newly introduced method is not discussed (nor reported): It is learning online, and requires sampling G*2 outputs per prompt and one pass through Comet, another neural net, and a pass to a language id model (potentially also neural?) to make one gradient update.
- Experiments outside of the sentence-aligned parallel data regime are missing: effectively still using both sides if trained for both directions. See questions below.
- Averaging across two different metrics with different effect sizes is not recommended (https://aclanthology.org/2024.acl-long.110/).
- The experimental setup is overall too narrow: evaluation metrics are those that are also in the reward (Comet models are still very similar in the end, even between reference-free and reference-based variants), so it's likely that this model is simply more optimized to those two metrics than any of the other compared models. It would be great to test the generalization to other evaluation metrics or what happens if one of the reward metrics is swapped by another one of its class, because it is widely known that overfitting the metric is easy in MT, and does not yield robust evaluation results (see e.g. https://aclanthology.org/2024.wmt-1.1.pdf).

**Questions:**

- Please fully define all variables introduced with GRPO in Eq 1. It is unclear how epsilon and G are set, and it is only specified in the appendix that the entire KL term is dropped due to setting beta to 0. The objective thus resembles MRT more - a common algorithm for RL for NMT (https://aclanthology.org/P16-1159.pdf).
- Comparison with SFT: How would SFT on the same amount of data (or RL with e.g. BLEU/XComet feedback with the same amount of samples, or offline rejection sampling with the proposed rewards) compare? How would SFT on forward translations fare (including the reverse direction)? This would also be reference-free and allow to disentangle the effects of input data vs reward modeling.
- Even though the data is taken apart (using each side as a monolingual corpus), the model ends up being trained on (multiway) parallel data because it is trained in both directions at once. I am very sure this has unintended benefits for the approach. Can you replicate the gains with truly monolingual, not crosslingually overlapping, data (also provided with every WMT task btw!)? Best for even lower resource languages? There are plenty of other languages to choose from in Flores 200 that are lower-resourced than the ones chosen.
- How is language mixing measured?
- What if RL training was done on top of an IFTed model? Since they already perform better, I would expect diminishing returns.
- Where does the data for Icelandic and Norwegian come from, also WMT dev/test sets?
- Which tokenizers are being used for sacrebleu? Please report the full sacrebleu signature and make sure to use the Chinese tokenizer when evaluating on Chinese on the target side (https://github.com/mjpost/sacrebleu/blob/a5425381d358b27555641bb0903bf4891775cb7b/sacrebleu/tokenizers/tokenizer_zh.py#L72) - likewise for other languages.
- Please report significance scores for very close comparisons, e.g. with sacrebleu (https://github.com/mjpost/sacrebleu/tree/a5425381d358b27555641bb0903bf4891775cb7b?tab=readme-ov-file#paired-significance-tests-for-multi-system-evaluation) for BLEU or comet-compare (https://unbabel.github.io/COMET/html/faqs.html#interpreting-scores).
- How does the more recent Tower+ (https://huggingface.co/Unbabel/Tower-Plus-9B ) compare? It was released in June, and is more competitive, so I would expect it is harder to beat.
- Please take language support into account when comparing performance across models. Some models simply don't support e.g. Icelandic - would you still consider it a fair comparison?

---

### Official Review · Reviewer_K15Q · 2025-11-01

**Soundness:** 2
**Presentation:** 3
**Contribution:** 2
**Rating:** 4
**Confidence:** 3

**Summary:**

The paper proposes Self-Trans, a reference-free reinforcement learning framework for machine translation (MT) that learns from round-trip consistency using only monolingual data. For each source sentence x, the model translates to y and back to \hat{x}; the reward combines a semantic term \(r_{\text{fwd}}\) (COMET-family QE on \((x,y)\)) and a \emph{lexical} term \(r_{\text{bwd}}\) (BLEU between \(\hat{x}\) and \(x\)), together with penalties that discourage trivial solutions such as source copying and language mixing. Training uses GRPO with simple prompting and a bidirectional setup. On \(\text{EN}\leftrightarrow\text{ZH}\) and several multilingual/low-resource directions, the approach improves over an SFT base and is competitive with strong baselines. The recipe is practical and reproducible, though evaluation later reuses the same metric families (BLEU/COMET).

**Strengths:**

A clear, runnable recipe for monolingual MT: the round-trip signal is decomposed into intuitive parts (semantic + lexical) and paired with sensible anti-cheating penalties.

Empirically, the approach delivers consistent gains across several directions and model sizes. The implementation details (prompting, GRPO, reward components) are straightforward, which increases reproducibility and potential impact.

**Weaknesses:**

Evaluation is not independent of the reward. The training signal and the reported test metrics come from the same metric families (BLEU and COMET variants). That invites reward hacking and makes it hard to tell whether quality actually improved. The paper should include an independent check—at least a small human SxS/MQM, or an LLM-as-a-judge evaluation—to validate real quality gains.

Positioning/baselines. The method is close in spirit to classic dual learning / unsupervised MT with round-trip objectives. There’s no head-to-head with strong modern implementations under matched data/compute, so the added value of the RL formulation isn’t fully isolated.

Over-optimization controls. Given the coupling between reward and evaluation, it’s surprising there is no reward-ensemble or transformed-reward baseline (now fairly standard to mitigate over-optimization).

Reporting could be stronger: add confidence intervals / permutation tests, and stress-tests with noisy or weaker monolingual data (where COMET/QE may be less reliable).

Also relevant is Mitigating Metric Bias in Minimum Bayes Risk Decoding (Kovacs, WMT 2024), which shows that MBR optimized with COMET/MetricX overestimates quality vs. humans—even when evaluated with related metrics—and recommends ensemble utility metrics; this strengthens our concern about using BLEU/COMET both as reward and as final metrics.

Missing related work (please cite with discussion and, where feasible, add baselines)

“Dual Learning for Machine Translation” — Di He, NeurIPS, 2016.

“Unsupervised Machine Translation Using Monolingual Corpora Only” — Guillaume Lample, ICLR, 2018.

“Helping or Herding? Reward Model Ensembles Mitigate but do not Eliminate Reward Hacking” — Jacob Eisenstein, COLM, 2024.

“Transforming and Combining Rewards for Aligning Large Language Models” — Zihan Wang, ICML, 2024.

(for independent MT evaluation) “Large Language Models Are State-of-the-Art Evaluators of Translation Quality” — Tom Kocmi, EAMT, 2023.

(on metric caveats) “Pitfalls and Outlooks in Using COMET” — Václav Zouhar, WMT, 2024.

**Questions:**

Independent evaluation: Can you add a small human SxS/MQM slice, or at least an LLM-as-judge evaluation, to decouple from BLEU/COMET?

Metric coupling: What is the Spearman/Pearson between your training reward and reported eval metrics?

Baselines: Can you include matched dual-learning / unsupervised MT baselines (modern back-translation/denoising recipes) under the same monolingual data and compute?

---

### Official Review · Reviewer_3JfE · 2025-11-03

**Soundness:** 2
**Presentation:** 3
**Contribution:** 2
**Rating:** 4
**Confidence:** 4

**Summary:**

This paper proposes Self-Trans, a novel machine translation framework that leverages dual learning and reinforcement learning to enhance translation performance using only monolingual data. It introduces a round-trip translation mechanism paired with a carefully designed reward architecture. To mitigate reward hacking, the authors propose several strategies, including composite evaluation metrics and anti-cheating penalties. Evaluated on standard benchmarks such as WMT and FLORES-200, Self-Trans achieves competitive results against strong baselines, demonstrating its promise for low-resource settings where annotated data is scarce.

**Strengths:**

1. The paper is clearly written and easy to follow, presenting a straightforward and intuitive idea.
2. It introduces carefully designed penalties to mitigate reward hacking in forward translation, and the results validate their effectiveness.
3. Experimental results demonstrate the superiority of the proposed framework, and the analyses shed light on the contributions of its key components.

**Weaknesses:**

1. This paper shares a core idea with dual learning [1], a well-established framework in machine translation. The failure to cite this foundational work is a significant oversight. Because the central concept closely resembles prior dual-learning approaches, the paper offers limited novel insights. While the design addressing reward hacking is reasonable, it does not appear particularly nontrivial or innovative.

2. Although round-trip translation avoids the need for parallel bilingual data, it still relies heavily on automatic metrics to evaluate forward-generation quality. The paper employs COMETkiwi, a reference-free metric, which inherently limits translation performance to its capabilities. In languages where COMETkiwi is unsupported or underperforming, the proposed method may fail to generalize effectively.

3. The comparison with the most relevant baseline of MT-R1-Zero is misleading. The paper incorrectly lists MT-R1-Zero as an 8B model; in fact, it is based on Qwen-2.5-7B. In contrast, this work uses Qwen3-8B as its base model, which is a stronger model that likely contributes significantly to performance gains. This discrepancy should be clearly acknowledged and discussed to ensure a fair assessment of relative improvements.

4. Reinforcement learning is performed on Qwen3-Base, which is relatively weak in translation capability compared to existing SFT models that already demonstrate strong translation performance. It would be significantly more convincing to evaluate the proposed RL approach on these powerful models.

[1]Xia et al, Dual Learning for Machine Translation.

**Questions:**

See weaknesses.

---

### Meta-Review · Area_Chair_YZHK · 2026-01-06

**Summary:**

All the reviewers shared severe concerns about the novelty of the method, the consistency between the motivation and the evaluation, and the validity of the empirical evaluation. Unfortunately, the authors did not provide any responses towards these concerns, leading them remaining outstanding. Therefore, I recommend a rejection.

**Reviewer Concerns:**

The authors did not provide any replies to the reviewers' concerns, all of which remain outstanding.

**Reviewer Scores:**

Reviewers will not change their score due to missing rebuttals.

---

### Decision · Program_Chairs · 2026-01-26

Reject